# Psychological Effects of Online-Based Mindfulness Programs during the COVID-19 Pandemic: A Systematic Review of Randomized Controlled Trials

**DOI:** 10.3390/ijerph19031624

**Published:** 2022-01-31

**Authors:** Young-Ran Yeun, Sang-Dol Kim

**Affiliations:** Department of Nursing, College of Health Science, Kangwon National University, 346 Hwangjo-gil, Dogye-eup, Samcheok-si 25949, Gangwon-do, Korea; yeunyr@kangwon.ac.kr

**Keywords:** COVID-19, online mindfulness, psychological problem, systematic review, randomized controlled trial

## Abstract

(1) Background: The COVID-19 outbreak has caused psychological problems worldwide. This review explored the psychological effects of online-based mindfulness programs during the COVID-19 pandemic. (2) Methods: This systematic review was guided by the Preferred Reporting Items for Systematic Reviews and Meta-Analyses guidelines. Randomized controlled trials that were published in the English language from 1 January 2020 to 31 May 2021 on online-based mindfulness programs for psychological problems due to the COVID-19 pandemic were searched in electronic databases. Quality assessment was conducted on the retrieved RCTs using the Cochrane risk of bias tool for RCTs. (3) Results: Six RCTs were included in this review. Quality appraisal of included RCTs ranged from 1 for low risk of bias to 5 for high risk of bias. There is evidence from the six RCTs that online-based mindfulness interventions may have favorable effects for reducing the levels of psychological problems, such as anxiety, depression, and stress. (4) Conclusions: Online-based mindfulness programs may be used as complementary interventions for clinical populations, healthy individuals, and healthcare workers with psychological problems due to the COVID-19 pandemic.

## 1. Introduction

The COVID-19 pandemic has changed human interaction from face-to-face to contactless [1,2,3]. Contactless human interaction is crucial to curbing the spread of COVID-19 [1,4]. However, contactless human interaction can trigger negative impacts on human psychosocial aspects [1,2,4] and lead to psychological problems, such as anxiety, depression, stress, and COVID-19 phobia [1,2,3,4,5]. These psychological problems may reduce an individuals’ ability to appropriately cope with reality [6,7,8]. This implies that timely interventions that aim to provide human psychological support should be implemented during the COVID-19 pandemic [6]. Previous studies have indicated that online-based psychological supportive therapies, such as online reassurance services [1], internet-based integrated intervention containing mindfulness [2], online cognitive behavioral intervention [3], online multimedia education [4], and deep breathing and stretching [5] can help fight psychological problems caused by the COVID-19 pandemic. These measures primarily help to modify individuals’ cognitive evaluation of suffering or stressful events; additionally, they assist individuals to discontinue their subsequent negative rumination triggered by the COVID-19 pandemic. According to prior theoretical and empirical evidence, mindfulness is a contributing factor toward individuals’ cognitive function modification and coping ability improvement [6,7,8]. Mindfulness involves focusing one’s awareness on the present moment without any judgment [6,9]; mindfulness programs may help discontinue the negative rumination related to COVID-19 by promoting cognitive behavior [3,6,7,8,9,10,11]. As mentioned above, timely interventions should be introduced to support the psychological stability of the general population during the global pandemic caused by COVID-19.

Online complementary and alternative therapies have been actively applied as psychological supportive approaches to combat the psychological crisis caused by COVID-19 [12,13]. Online psychological support interventions are considered because of their contactless nature. A prior review has suggested that internet-based psychological support interventions qualify as complementary and alternative therapy to face-to-face therapy [14]. Furthermore, the COVID-19 preventative measure of social distancing restrains conventional psychological therapeutic approaches; therefore, there is a need for online-based psychological interventions [2,4]. Previous studies have emphasized that online-based psychological interventions provide convenient access to psychological support while observing COVID-19 preventative guidelines [2,3,4,15]. According to previous studies, online-based mindfulness programs include meditation [16], imagery [3,6], tapping [15], and various types of relaxation techniques [2,4]. These programs have been applied to relieve psychological problems during the COVID-19 pandemic.

Nevertheless, evidence-based interventions focused on persons with psychological problems due to the COVID-19 outbreak are still not enough [2,13]. Furthermore, to provide accessible and evidence-based interventions to persons with psychological problems due to sudden global or public health crises, such as the COVID-19 pandemic, more studies that apply a rigorous research methodology should be preceded [1,2]. A recent rapid review study investigated the literature related to the psychological impacts of COVID-19 and how to mitigate them. The study pointed out that formal quality appraisal has not been conducted in the foregoing literature. However, no randomized controlled trials (RCTs) were included in this review study [1]. Another review study considered the literature published until 16 July 2020 and recommended that further rigorous studies using different complementary and alternative medicine approaches for psychological health in COVID-19 patients be undertaken [13]. Overall, according to previous studies, it is speculated that online-based mindfulness interventions will help alleviate participants’ psychological problems during the COVID-19 pandemic. Furthermore, it is considered necessary to evaluate whether the effects of online-based mindfulness intervention were derived through a study that applied a rigorous research methodology. Based on the above evidence, we set the hypothesis as follows. During the COVID-19 pandemic, online-based mindfulness programs by RCTs will be effective in reducing participants’ levels of psychological problems. In these contexts, our study aimed to assess the psychological effects of online-based mindfulness programs through a systematic literature review targeting peer-reviewed and RCTs published from 1 January 2020 to 31 May 2021.

## 2. Materials and Methods

### 2.1. Literature Search

This systematic review was guided by the Preferred Reporting Items for Systematic Reviews and Meta-Analyses (PRISMA) guidelines [17]. Electronic databases, including the Cochrane Library, Embase, ProQuest (PsycINFO), PubMed, and Web of Science, were searched to identify RCTs that have reported the effects of online-based mindfulness programs on psychological problems related to COVID-19 published in the English language from 1 January 2020 to 31 May 2021. We searched for the terms “online mindfulness” AND “COVID-19” OR “psychological” AND “COVID-19”, and for the search filter, the randomized controlled trial was limited. All the articles identified during this search that met the selection criteria highlighted in the next section were reviewed. The footnote chasing method was used for screening additional articles related to target search terms [18].

### 2.2. Selection Criteria

The inclusion criteria were evaluated using the PICO (population, intervention, comparison, and outcome) elements in RCTs [19]. For population, RCTs targeting human beings with psychological problems due to the COVID-19 pandemic were included regardless of gender, age, and country of participants. For intervention, RCTs comparing the use of online-based mindfulness intervention to relieve psychological problems, such as anxiety, depression and stress, due to the COVID-19 pandemic were included. The scope of online-based mindfulness intervention included contactless approaches, such as online platform, app, weblinks, internet, Zoom, and online portal by mobile phone or computer. For comparison, any RCTs comparing online-based mindfulness intervention for reliving psychological problems due to the COVID-19 pandemic for at least one psychological variable, such as anxiety, depression, and stress, versus the number of mindfulness programs were included. For outcomes, RCTs that assessed psychological variables, such as anxiety, depression, and stress, were included [1,13,14].

### 2.3. Data Extraction

This review included extracted data on the characteristics of participants (sample size, mean age, and dropout rate and percentage), interventions (online-based mindfulness programs, delivery method, duration, and interventionist), outcome measures (psychological variables such as anxiety, depression, and stress), main results (the mean difference between online-based mindfulness intervention group vs. control group), adverse events (safety), and limitations. The analysis excluded nonrandomized controlled trials, full texts, and target intervention in line with the Consolidated Standards of Reporting Trials criteria [20].

### 2.4. Quality Assessment

Quality appraisal was conducted by the Cochrane risk of bias tool for RCTs [21] following six domains and seven items of the tool, adjusted: selection bias (random allocation and allocation concealment), performance bias (blinding of participants and researchers), detection bias (blinding of outcome assessment), attrition bias (incomplete outcome data), reporting bias (selective reporting), and other bias. The risk of bias for each item was rated as “low”, “high”, or “unclear”. The overall risk of bias for each RCT was assessed at three levels (A, B, and C): level A if they had a low risk of bias for all assessed domains, level B if they had a low or unclear risk of bias for all assessed domains, and level C if they had a high risk of bias for one or more domains [19].

### 2.5. Data Synthesis

A systematic review was performed because the psychological variables and types of online-based mindfulness programs of the included RCTs were heterogeneous. In addition, no meta-analysis was performed because data, such as mean, standard deviation (SD), and effect size, were not available.

## 3. Results

### 3.1. Study Description

Overall, 121 RCTs related to the search terms of this review were screened. Among these, 36 were identified from the Cochrane Library, 9 from Embase, 40 from ProQuest (PsycINFO), 3 from PubMed, and 33 from Web of Science. Eight RCTs were excluded because they had duplicates. For the remaining 113 RCTs, 61 were excluded because they had no target intervention and no clinical trials, 50 were excluded for having no study protocols, no online-based intervention, no RCTs, and not being within the COVID-19 outbreak period. Four additional RCTs were included by the footnote chasing method. Finally, six RCTs were selected for this review. The literature retrieval process is depicted in Figure 1. The characteristics of the included RCTs are presented in Table 1.

### 3.2. Quality Assessment

Quality assessment for the six included RCTs was performed independently by two authors. The quality appraisal resulted in a 97.6% rater agreement across domains and items of the Cochrane risk of bias tool for RCTs. Disagreements between authors were discussed in depth to reach a consensus. Quality appraisals for the six included RCTs are described in Table 2. Of the six RCTs, five were identified as quality level C, and the remaining one as quality level A.

### 3.3. Participants and Settings

The six RCTs originated from China, Germany, Iran, Sweden, Turkey, and the U.S.A. (Table 1). Participants were patients with COVID-19, nurses and residents caring for and treating COVID-19 patients, obstetrics and gynecology patients during COVID-19, and the general population with dysfunctional worry due to the COVID-19 pandemic. Sample sizes of the six RCTs ranged from 26 to 670 participants and totaled 976 participants. Of the 976 participants, 93 dropped out, with a dropout rate of 9.5%. Participants’ ages ranged within 18–79 years, except for three RCTs that did not indicate the age range of the participants.

### 3.4. Design and Intervention

The online-based mindfulness programs applied in the six RCTs included in this review were meditation in one RCT [16], tapping in one RCT [15], imagery in two RCTs [3,14], and integrated relaxation techniques in two RCTs [2,4] (Table 1). The duration of these programs ranged from 20 min for a single session to 2 weeks in three RCTs, and 3 and 4 weeks in the remaining two RCTs. The practice time per session ranged from 10 to 120 min. Online-based mindfulness programs were performed under the authors of RCTs in four RCTs, psychologists in one RCT, with no information for the remaining one RCT.

### 3.5. Main Outcomes

Anxiety, depression, and stress were accessed in the included RCTs for this review (Table 1). Anxiety and depression were assessed in five RCTs, and a significant decrease in anxiety and depression levels was observed in the online mindfulness groups than in the control group. Stress was assessed in three RCTs, and the stress or distress levels were significantly decreased in the online mindfulness groups than in the control groups. These findings support our hypothesis.

## 4. Discussion

This review sought to assess the effects of online-based mindfulness programs for psychological problems, such as anxiety, depression, and stress, due to the COVID-19 pandemic. Data were reported descriptively due to the heterogeneous characteristics of RCTs included in this review and insufficient statistical data for meta-analysis. We found that six RCTs included in our review comprised two RCTs targeting patients with COVID-19 [2,4], one RCT targeting obstetrics and gynecology patients during the COVID-19 pandemic [16], one RCT targeting the general population with dysfunctional worry due to COVID-19 [3], and two RCTs targeting healthcare workers treating or caring for COVID-19 patients [6,15]. Our review is of great significance because it evaluates the effect of online-based mindfulness programs on psychological problems from not only clinical populations with COVID-19 or obstetrics and gynecology patients, but also healthy individuals including the general population or healthcare workers, such as nurses and residents. Moreover, the online-based mindfulness interventions and location of their publication included in this review were originated in Asia [4,6,15], America [16], and Europe [2,3]. These findings can be considered surprising and meaningful data that reflect the current situation of the COVID-19 pandemic. If the efficacy and safety of online-based mindfulness programs are guaranteed through further investigations due to a few previous studies, it can be estimated that they can be applied to various subjects and locations as complementary and alternative therapies for controlling the urgent psychological crises during the COVID-19 pandemic [1,2]. Different types of online mindfulness interventions, such as meditation [16], tapping [15], imagery [3,6], and relaxation techniques [2,4], were applied in RCTs included in this review. It is presumed that the standardization of the online-based mindfulness programs’ usage to provide appropriate prescriptions or dissemination according to target consumers who have psychological problems in the face of future public health crises or sudden societal changes is required through additional studies [3,4,15].

The effects of online-based mindfulness programs on psychological problems related to COVID-19 are as follows: the online-based relaxation practices showed a significant decline in the levels of anxiety, depression, and stress in the trained group compared to the control group, which can be evaluated as evidence for psychological impacts control in COVID-19 patients [2,4]; however, because the effects of the relaxation techniques were observed in only two RCTs, further investigations with the targeted COVID-19 populations are needed. Additionally, online self-guided practice showed a significant decline in the levels of anxiety and depression in the trained group compared to the control group, which can be evaluated as evidence for a reduction in the anxiety and depression levels from the general population with dysfunctional worry related to the COVID-19 pandemic [3]. Other imagery programs showed a significant decline in the level of anxiety and stress in the trained group compared to the control group, which can be evaluated as evidence for a reduction in anxiety and depression levels in residents treating COVID-19 patients [6]. In addition, mindfulness meditation practice showed a significant decline in the levels of anxiety, depression, and stress in the trained group compared to the control group, which can be evaluated as evidence for a reduction in the anxiety, depression, and stress levels in obstetrics and gynecology patients, but this was observed in only one RCT [16]. Interestingly, tapping practice as an emotional freedom technique showed a significant decline in the level of anxiety and stress in the trained group compared to the control group, which can be evaluated as evidence for a reduction in anxiety and stress levels in nurses caring for COVID-19 patients [15]. As described above, it is noteworthy the findings concerning reductions in anxiety, depression, and stress levels in the trained group compared to the control group, in various participants, different intervention locations, and different types of online-based mindfulness programs. These findings will be meaningful data for the psychological health regimen or guidelines not only for COVID-19 patients, but also for the general population with psychologically unstable states due to the COVID-19 pandemic, and healthcare workers caring or treating COVID-19 patients globally [3]. Whether it is caused by COVID-19 itself or quarantine, it is a well-known fact that the COVID-19 pandemic has acted as a threat to human psychological health. Until COVID-19 ends in our daily lives, it is thought that continuous interventions for psychological health should be developed, and appropriate interventions should be provided to target clients or individuals [1,2,13].

The quality rating of the six RCTs included in our review had a high risk of bias in the five existing RCTs, except for a low risk of bias in one RCT. Five of the six RCTs were not double-blinded, owing to all the blinding of participants and researchers. These findings stem from fact that no double-blinding was the most crucial flaw in existing RCTs’ included online-based intervention, as is often noted in RCTs for face-to-face intervention [1,14], wherein no double-blinding is believed to be due to the authors in the RCTs included in this review having contacted their participants for informed consent and proceeded with the online-based intervention due to the quarantine necessitated by the COVID-19 pandemic and the insufficiently guaranteed intervention duration during the COVID-19 pandemic [3,4,15,16]. Blinding of the outcome assessment item of detection bias was adequate in the existing RCTs except for three RCTs. Fortunately, in contrast, the random sequence generation, and the allocation concealment items of selection bias by using online methods, the selective reporting item of reporting bias were adequate in all the existing RCTs. Incomplete outcome data items of attrition bias were adequate in all the existing RCTs, except one RCT that included dropped-out participants’ data due to small sample sizes. In conclusion, despite the RCTs that conducted online-based intervention, the overall risk of bias for each RCT can be summarized as a high risk of bias in five RCTs, except for one of the six RCTs, due to major drawbacks of no double blinding.

The strengths of our review are as follows. To the best of our knowledge, this is the first comprehensive review involving RCTs of the effectiveness of online-based mindfulness intervention for individuals who have been psychologically affected by the COVID-19 pandemic. Compared to a previous rapid systematic review that included studies on psychological impacts due to quarantine regardless of the year of publication and style of studies [1] and another review on complementary and alternative medicine approaches involved in only COVID-19 patients [13], our review differs from the two previous reviews in that the RCTs included psychological variables, such as anxiety, depression, and stress due to quarantine due to COVID-19 or the disease. Additionally, the RCTs in our review included east-west location, COVID-19 pandemic period, and online platform use for mindfulness intervention that was extended and applied to COVID-19 patients, healthy individuals, and healthcare workers. Based on this scientific evidence, online-based mindfulness interventions may help to improve psychological negative impacts and fight against the COVID-19 pandemic.

However, this review has the following limitations. Firstly, there was the generalization problem due to the small sample size in the existing five RCTs, except for one RCT [3]. Secondly, a notable limitation is that the heterogeneity of our sample, which included COVID-19 patients, healthy individuals, and healthcare workers, limits the generalizability of our findings. Future research is encouraged to analyze the effects of online-based mindfulness intervention on target population groups. Thirdly, there was a performance bias owing to double blinding due to the involvement activities of the study authors and personnel for the participants’ acceptability and feasibility during the COVID-19 pandemic situation, applying contactless intervention. Fourthly, there were measures of the short-term effects of online mindfulness programs. Statistical variables that can calculate the effect size of the study results were not presented. It is necessary to check the effect size through meta-analysis by integrating the research results of each RCT. Another weakness is the fact that the effect was identified not by objective physiological measurement data, but by the subjective evaluation of the research participants. Therefore, it is necessary to compare them with each other by securing physiological data in a repeat study or follow-up in the future. Moreover, there was no mention of adverse events in the six RCTs compared to the high dropout rate. Although the efficacy of the research results is important even if it is conducted under urgent circumstances due to the COVID-19 pandemic, a strict research methodology must be adhered to for the safety of the subjects. Furthermore, it is necessary to actively develop online platforms for safe and effective online mindfulness intervention in the future and to open them to the subjects. Now, in the application of complementary therapies, including mindfulness, it is necessary to pay more attention to the enhancement of security for accessibility and personal information exposure risk rather than efforts to determine the place of intervention. Taken together, as pointed out in previous review studies [1,7], these reviewed RCTs also have the vulnerability of research methodologies, such as a small sample size, few RCTs, controlled assignment studies with no treatment, short-term intervention, no long-term effects, and reliance on self-reported data in RCTs for reliving psychological problems during the COVID-19 outbreak. To overcome the limitations described above, further investigations for high-quality RCTs according to rigorous methodologies are required.

## 5. Conclusions

In conclusion, the findings from the six RCTs suggest that online-based mindfulness programs may be used as complementary interventions for clinical populations, healthy individuals, and healthcare workers with psychological problems due to the COVID-19 pandemic.

## Figures and Tables

**Figure 1 ijerph-19-01624-f001:**
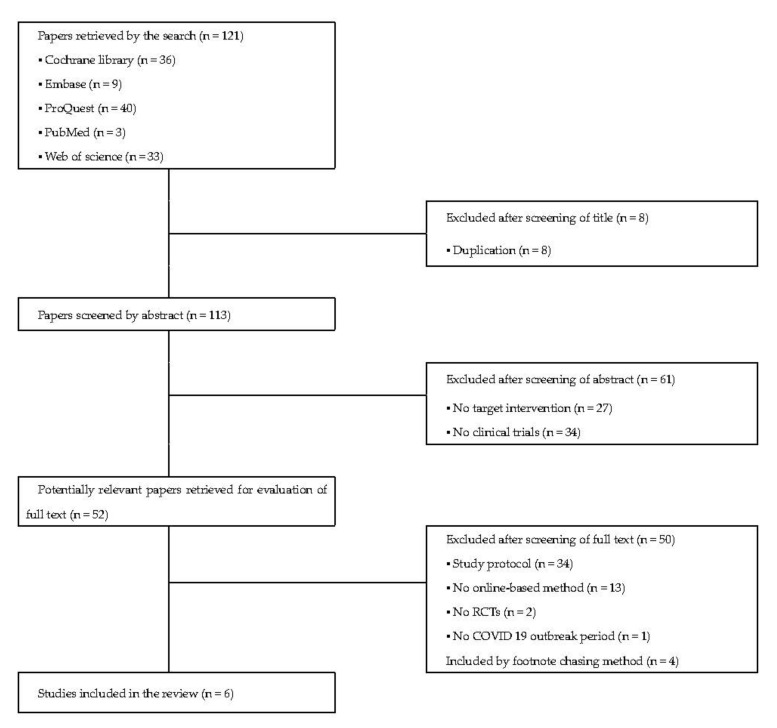
Flowchart of included studies through the literature searches.

**Table 1 ijerph-19-01624-t001:** Characteristics of the included randomized controlled trials.

Author, Year, Country	Participants	Interventions		Outcome - Measures		Results		Adverse - Events	Limitations
		Experimental Group	Control Group		MD	95% CI	*p*-Value		
	Population Sample size: n (EG, CG) Mean age: years (range) Drop out: n (%)	Interventions Delivery method Duration Interventionist							
Diner and Inangil 2021, Turkey	Nurses caring for COVID-19 patients 72 (35, 37) 33.5 (NR) 0 (0.0)	EFT (Tapping) Survey Monkey and Zoom 20 min/single session First author certified in EFT	Stay in calm and tranquil environment for 15 min after surveying	STAI SUD	NR NR	−35.18 to −29.16−5.02 to −3.89	*p* < 0.001 *p* < 0.001	NR	Small sample size No biological indicators such as cortisol levels & genes expression related to stress
Shaygan et al., 2021, Iran	Patients with COVID-19 50 (27, 23) 36.8 (31–40) 2 (4.0)	OMPI WhatsApp Two weeks/60 min per session First author	Psychological counseling if needed	PSS	−0.77	−12.8 to −1.78	*p* = 0.01	NR	Small sample size Lack of a long-term follow up Self-reported adherence
Smith et al., 2021, U.S.A.	OB & GY patients 101 (50, 51) 35.6/36.8 (NR) 11 (10.9)	Mindfulness meditation MobileApp Calm 30 days/10 min per day Corresponding author	Standard care	PSS Anxiety Depression	4.28 1.89 1.36	1.68 to 6.88 0.06 to 3.72 0.04 to 2.68	*p* = 0.002 *p* = 0.04 *p* = 0.04	NR	Small sample size Heterogeneous population Self-reported outcome measures
Zhang et al., 2021, China	Chinese residents 57 (29, 28) 50.1 (NR) 6 (10.5)	MBSR Online platform (WeChat) 14 days/three times per day 2 h per session Psychologist	Waitlist control	Anxiety Depression	−8.09 −7.16	NR NR	*p* < 0.001 *p* < 0.001	NR	Small sample size Use of waitlist control Common methods bias No long-term effects
Wahlund et al., 2020, Sweden	General population with dysfunctional worry related to COVID-19 670 (335, 335) 45/47 (18–81, 19–79) 71 (10.6)	Self-guided program Online platform (study website) 3 weeks Study personnel	Free to seek other kinds of help if needed	GAD-7 MADRS-S	0.74 0.38	0.58 to 0.90 0.22 to 0.55	*p* < 0.001 *p* < 0.001	NR	Threat to the external validity Self-reported outcome measures
Wei et al., 2020, Germany	Patients with COVID-19 26 (13, 13) 40.8/48.5 (18–65) 3 (8.7)	IBII Online platform 2 weeks/session for daily 50 min per session NR	Supportive care 2 weeks/daily	17-HAMA HAMD	NR NR	NR NR	*p* = 0.001 *p* < 0.001	NR	Small sample size A risk of bias for blinding Study period of two weeks

EFP, emotional freedom techniques (tapping); IBII, internet-based integrated intervention (breath relaxation, mindfulness, refuge skills, bufferly hug); MBSR, mindfulness-based stress reduction program (breathing, focus on present moment, attending to the sensations, noticing breath, and being mindful); NR, not reported; OMPI, online multimedia psychoeducational intervention (various types of relaxation techniques); OB & GY, Obstetrics and Gynecology. CI, confidence interval (lower to upper) of the difference; GAD-7, generalized anxiety disorder 7-item scale; HAMA, Hamilton anxiety scale; HAMD, Hamilton depression scale; MADRS-S, Montgomery Åsberg depression rating scale—self report; NR, not reported; PSS, perceived stress scale; MD, mean difference between the two groups; STAI, state-trait anxiety inventory; SUD, subjective units of distress scale.

**Table 2 ijerph-19-01624-t002:** Quality appraisal of all included RCTs.

Authors, Year, Country	Selection Bias		Performance Bias	Detection Bias	Attrition Bias	Reporting Bias	Other Bias	Overall Risk of Bias
	Random Sequence Generation	Allocation Concealment	Blinding of Participants/Researchers	Blinding of Outcome Assessment	Incomplete Outcome Data	Selective Reporting		
Diner and Inangil, 2021, Turkey	+	+	−/−	+	+	+	+	C
Shaygan et al., 2021, Iran	+	+	+/+	+	+	+	+	A
Smith et al., 2021, U.S.A.	+	+	−/−	?	+	+	+	C
Zhang et al., 2021, China	+	+	−/−	+	+	+	+	C
Wahlund et al., 2020, Sweden	+	+	−/−	−	+	+	+	C
Wei et al., 2020, Germany	+	+	−/−	−	−	+	+	C

+, low risk of bias; ?, unclear risk of bias; −, high risk of bias; A, a low risk of bias for all assessed domains; B, a low or unclear risk of bias for all assessed domains; C, a high risk of bias for one or more assessed domains.

## Data Availability

The data presented in this study are available on the electronic journal website.

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
