# Peer review of "Psychological Effects of Online-Based Mindfulness Programs during the COVID-19 Pandemic: A Systematic Review of Randomized Controlled Trials"

_ijerph, 2022, doi:10.3390/ijerph19031624_

Round 1

Reviewer 1 Report

Comment 1: Figure 1 has to be modified to accurately meet the PRISMA template for PRISMA flowcharts.  Every step including keywords, number duplicates, exclusion/inclusion criteria should be described and reported on the flowchart.

Comment 2: A limitation is that the populations of the RCT studies are dissimilar which does not permit unified conclusions. This limitation has to be reported and discussed.

Comment 3: Another limitation is that there are only 6 RCT studies, which indicates that this systematic review may has been attempted prematurely. I think that this Systematic Review should be accompanied by a meta-analysis (effect sizes and statistical power to be evaluated). So, when the RCTs quantity will be increased, then, to conduct a proper systematic review and meta-analysis.

Comment 4: Also, the sample sizes of the studies are really small. The authors reported that the data (SDs, means, and effect sizes) are not available. This is another significant limitation. The authors should attempt to implement a Synthesis without meta-analysis (SWiM) (see https://www.bmj.com/content/368/bmj.l6890 ).  However, I am really concerned that while the authors opted for reviewing the RCT studies there are not effect sizes, which is the key for evaluating the quality and impact of the interventions.  So, I am not sure if the SWiM would be informative for studies with RCTs. Given that the number of studies is really small (only 6 studies), I think that the authors should contact all the authors of these 6 studies and request access to the means, SDs, and effect sizes.  

Author Response

The authors’ responses to your review are sent as a file.

With best regards

Prof. Kim 

Reviewer 2 Report

Dear authors, 

The paper review is about the impact of mindfulness in a context pandemic SARS-CoV-2, however, the authors have (in my personal and professional opinion) a little issue to resolve: When I use the term COVID-19 and when I use the Term SARS-CoV-2. All of us know this is a different term (in academic science). In fact the term COVID-19 was use to describe everything about the pandemic, and this is really BAD for all (scientists and common persons) because isn't correct.

In yours study you are three different situations: persons with COVID-19; persons treated other persons with COVID-19; and the persons with a fear to contrains the virus SARS-COV-2 and maybe will can be diease. You need to resolve them.

The article is very good, but if you can clarify these aspects better, they would bring more robustness to the work and perhaps contribute/raise awareness for the correct use of the terms SARS-CoV-2 and COVID-19.

Other limitations of your work is the impact of mindfulness in the different samples (with COVID; fear of contract SARS-CoV-2 (associated to COVID-19 disease"; fear of COVID (intensive care unit); and others). This limitations can be a futures studies.

Congratulations!

Author Response

(The authors gave the same response as above.)

Reviewer 3 Report

This a systematic review of the literature on virtual mindfulness therapies during the COVID-19 pandemic. This is an interesting and timely study; however, the aims and hypotheses of this study are poorly described. Here are my concerns/suggestions.

  • “The COVID-19 pandemic has changed human interaction from face-to-face to untact…” No definition has been provided for “untact”. Please replace “untact” with an English word throughout the manuscript.
  • “Previous studies have indicated that online-based psychological supportive therapies can help fight the psychological crisis caused by the COVID-19 pandemic [1–5].” What do you mean by supportive therapies? Does it include CBT? If not, some of the cited references are from CBT studies. Please clarify in the manuscript.
  • “In these contexts, our study aimed to assess the scientific evidence and psychological effects of online-based mindfulness programs through a systematic literature review targeting peer-reviewed and RCTs published from January 1, 2020 to May 31, 2021.” The aim of this study is poorly described. What were your hypotheses? In it is not clear what questions this systematic review is trying to answer.
  • The introduction provides minimal background info on the objective of this study, which is virtual mindfulness therapies during the COVID-19 pandemic. Please revise this section and provide more background info on virtual mindfulness therapies.
  • “We searched for the terms “online mindfulness” AND “COVID-19” OR “psychological” AND “COVID-19”,” I believe the current search terms are not comprehensive. “online mindfulness” is very specific and it is not sensitive enough.
  • Regarding “quality assessment”, it is important to know if one of the authors or both of them did this. Ideally, more than one person should do it, and the authors need to include the results of their interrater reliability analysis.
  • The manuscript would benefit from English editing.

Author Response

(The authors gave the same response as above.)

Round 2

Reviewer 1 Report

The authors improved their manuscript, which now has an adequate quality for publication.

Author Response

The manuscript has been rechecked and the necessary changes have been made in accordance with the reviewers’ suggestions. Thank you for your consideration. 

Reviewer 3 Report

The authors have addressed some of my comments in the first round of review.

The authors need to clarify what question(s) this systematic review is trying to answer. These questions should be clearly stated in the introduction.

In Table 1, the authors need to replace “p values” with 95% confidence intervals along with the numerical difference in the scores (for example numerical difference in anxiety scale’s score).

“This may be the first comprehensive review of RCTs’ effectiveness of online-based mindfulness intervention for psychological impacts during the COVID-19 pandemic.” This statement should be revised.

“In conclusion, findings from the six RCTs suggest that online-based mindfulness programs may be used as complementary interventions for clinical populations, healthy individuals, and healthcare workers with psychological problems due to the COVID-19 pandemic.” The authors should acknowledge that the literature on this subject is very heterogeneous and that these findings need to be confirmed by future studies.

This manuscript would benefit from English editing by a subject-matter expert.

Author Response

The manuscript has been rechecked and the necessary changes have been made in accordance with the reviewers’ suggestions. The responses to all comments have been prepared and attached via a file. Thank you for your consideration. I look forward to hearing from you.
